# The Role of the Andalusian Institute for Agrarian Reform (IARA) in Irrigation Expansion: The Case of the Chanza Irrigation Project (Huelva, Spain)

José Manuel Jurado Almonte [1,*] and José Díaz Diego [2,*]

1    Department of History, Geography and Anthropology, University of Huelva, 21071 Huelva, Spain
2    Department of Social Anthropology, Basic Psychology and Public Health, Pablo de Olavide University, 41013 Seville, Spain
*    Correspondence: jurado@uhu.es (J.M.J.A.); jdiadie@upo.es (J.D.D.)

**Abstract:** In the last two decades of the 20th century, irrigation in Andalusia experienced a historic expansion as a result of the transfer of political powers from the State to regional authorities and, thanks to its application in Andalusia, to pass, among other measures, an agrarian reform bill and the subsequent development of new irrigation infrastructures. Against this background, our objective was to determine the role of the body responsible for implementing agrarian reform, i.e., the Institute for Agrarian Reform (Instituto Andaluz de Reforma Agraria, IARA), by converting drylands into irrigation lands, with a special focus on one of the region's most vibrant agro-economically transformed areas: the irrigable area of the River Chanza (the west coast of the province of Huelva). To conduct the study, we have applied the historical method and content analysis to technical, legal and agro-statistical documentation from the entire active period of the IARA (1984–2011). The results highlight the leading role played by the IARA in extending irrigation land in agrarian-reform priority areas, as well as in developing irrigation infrastructures in the areas that the State defined as strategic in Andalusia, such as the 17,200 hectares of irrigated land of the Chanza Irrigation Project. We conclude that the radical agro-productive transformation of the coast of Huelva in recent decades is mainly due to the availability of water for irrigation, which the IARA was primarily responsible for planning and executing.

**Keywords:** IARA; agrarian reforms; irrigation schemes; water; agricultural landscapes

## 1. The IARA and the New Irrigated Areas in Andalusia. Policy History

The unbalanced distribution of land ownership and low agricultural productivity has been a long-standing historical problem in Andalusia. From the mid-18th century until well into the 20th century, there have been several unsuccessful proposals and attempts to correct this situation. These include the Conde de Aranda agricultural reform provisions (1766–1770), the Pablo de Olavide land reform project report (1777), the "Caroline" municipal land allocations (1767–1854), the disentailment of communal land and mortmain land (1798–1924), the protectionist provisions of the Institute for Social Reforms (1903–1924), the Augusto Besada Agrarian Reform Law (1907) and the initiatives of the Central Board of Colonisation and Interior Repopulation (1907–1925), the José Canalejas Agrarian Reform Bill (1911), the Dato-Lizárraga Agrarian Reform Bill (1921), the agrarian decrees of Largo Caballero and Fernando de Los Ríos (1931), or the Agrarian Reform Law of the Second Republic (1932), which was cut short by the Spanish Civil War and the subsequent Franco dictatorship [1–6].

During Franco's dictatorship, the National Institute for Colonisation [7] (Instituto Nacional de Colonización, hereinafter INC) was established in 1939. It aimed to alleviate the socio-economic issues of the rural areas in post-Civil War Spain, promoting irrigation and settlement, albeit by means of very specific actions, which in Andalusia did not substantially

change the precariousness of the day labourer sector. Later on, the dictatorship attempted to boost modernisation in unused agricultural areas, mainly under the public domain, concentrating all responsibilities on agrarian structures, (namely the National Institute for Colonisation, the General Directorate for Colonisation and the National Service for Land Consolidation and Rural Development) in a new body—the National Institute for Agrarian Reform and Development [8] (Instituto Nacional de Reforma y Desarrollo Agrario, hereinafter, IRYDA)—promoting a new Agrarian Reform and Development Law (Decree 118/1973 [9]). This did not improve, neither in ambition nor in audacity, the 1932 Republic initiative [10]), but at least it did not waver from the conviction that it was necessary to hand over land to farmers, which the IRYDA carried out under administrative concession and subsequent ownership.

During the transition to democracy, support was given to the Substantially Improvable Holdings Law (Ley de Fincas Manifiestamente Mejorables [11]), which introduced the possibility of forced expropriation for the new political order of unused land as an instrument in the management and modernisation of agricultural holdings. This law's implementation fell under the responsibility of the IRYDA, which sublet the expropriated holdings.

Although the legislative framework enabled progress to be made in terms of restructuring the holdings, it was not put into practice, thus aggravating the problem of land in regions such as Andalusia or Extremadura, which at the time were highly dependent on an agrarian sector that, as a result of the progressive mechanisation of the countryside and the difficult working conditions it offered to those who remained in the countryside, was driving out the labour force. Unemployment and precariousness had transformed Andalusian agriculture into a "population expulsion mechanism" [12].

With the passing of the Spanish Constitution (1978) and the Statute of Autonomy of Andalusia (1981), this region took on the responsibilities previously assigned to the IRYDA, pursuant to Royal Decree No 1129/1984, of 4 April 1984, on the transfer of functions and services from the General State Administration to the Autonomous Community of Andalusia in the field of agricultural reform and development [13]. The new sphere of competence enabled the Andalusian political forces to promote autonomous legislation designed to meet the socio-economic needs of Andalusia, and also those of the workers and farmers. In this context, the political powers incorporated the administration's obligations to carry out agrarian reform into the 1979 statutory agreements. With its approval by referendum in 1981, the Statute of Autonomy of Andalusia made said political negotiation prescriptive. As a result, the Agrarian Reform Draft Bill, finally approved in 1984, was submitted to Parliament [14].

This agrarian reform became a political milestone given its uniqueness in a European context. In the Spain of the Transition to Democracy, Extremadura was the only region to raise the need to reform its agrarian structures, although its politicians avoided submitting a special law to their regional parliament, resolving to tentatively pursue the goal of reform through sub-sectoral regulations. In neighbouring Portugal, after the Carnation Revolution of 1974, agrarian reform had already stopped progressing in its implementation some years previously and was about to be liquidated by the State. And somewhat further afield, in Eastern Europe, the difficulties of the collectivised agriculture of the Soviet Union and its satellite countries had become a symbol of collectivisation inefficiency for the leaders of the European Economic Community; thus, dispensing with any support to collectives in the Common Agricultural Policy schemes and incentives. The Andalusian-Agrarian Reform Bill of 1984, which was passed when the sign of the times seemed to point in the opposite direction, has been labelled as anachronistic, contradictory and dispensable [15,16], although the mixed results of the reform can be better explained by political scepticism than by extemporaneous difficulties. Nevertheless, under the protection of reformist regulations, significant transformations were introduced, especially in the area of irrigation, as outlined below.

The Agrarian Reform Law 8/1984 (Ley de Reforma Agraria de Andalucía, hereinafter, LRAA) created, as the main instrument for the achievement of reformist goals, the Andalusian Institute for Agrarian Reform (hereinafter, IARA), a self-governing body attached

to the then Regional Ministry of Agriculture and Fisheries (Consejería de Agricultura y Pesca, hereinafter, CAP). The LRAA was adopted to address a significant underutilisation of land, and it also contained measures empowering the IARA to expropriate and purchase holdings to distribute among farmers. The aim was to satisfy a social demand concerning the issue of land with deep historical and socio-cultural roots.

In the framework of this reform momentum, little is known about the leading role played by the IARA in converting many agricultural areas to irrigation land, especially in "agrarian reform areas". In fact, as we will present below, the extension of irrigation land managed by the IARA could be considered its main legacy, even more so than work carried out in the area of restructuring land ownership and its productive modernisation.

Since its foundations, the LRAA used the extension of irrigation areas as a tool for modernisation. In Chapter V, titled "The conversion of large areas in the interest of the Autonomous Community", measures were included to declare priority areas whose irrigation expansion would be strategic for socio-economic development (Art. 42 and 43 of the LRAA) to be of general interest. To this end, the Ministry of Agriculture and Fisheries, at the request of the IARA, submitted the initiative to the Governing Council for its opinion, which, if approved, would publish it as a Decree. It then became the responsibility of the IARA to design and process the General Transformation Plan for the area with the mechanisms to extend irrigation in the selected perimeter, including the seizure of property and rights whose expropriation was necessary for the conversion of the area (Art. 42.4 of the LRAA).

Indeed, the activity of the IARA in terms of strengthening and extending irrigation areas in Andalusia was frenetic since the approval of the regulation for the implementation of agrarian reform (Decree 276/1984, of 30 October [17], replaced by Decree 402/1986, of 30 December [18]). Between 1986 and 1991, 15 decrees declaring areas to be "in the interest of the Autonomous Community" were enacted: 13 on irrigation and two on general measures, with a particular focus on irrigation (Table 1). With each decree, the IARA was empowered to promote the General Conversion Scheme for the area and all the initiatives and works necessary to achieve it, specifically those relating: (1) the creation of infrastructure for the transport and use of water in the converted area, (2) the creation of suitable holding units and the allocation to their beneficiaries where applicable and (3) the remaining actions to be set out in the Conversion Scheme (Art. 43.1 of the LRAA).

**Table 1.** Declarations of general interest regarding the Autonomous Community of Andalusia in terms of irrigation conversion.

| Decree | Year | Irrigable Area (ha) |
|---|---|---|
| Conversion into irrigated land of the Los Humosos area (Seville) | 1986 | 4272 |
| Irrigable area of the final stretch of the Guadalquivir in the municipality of Lebrija in the province of Seville, and Trebujena in the province of Cádiz | 1986 | 3642 |
| Conversion into irrigated land of the irrigable area of Donadío-cota 400 (Úbeda and Baeza, Jaén) | 1986 | 3800 |
| Conversion into irrigated land of the irrigable area of Valdemaría (Palos de la Frontera and Moguer, Huelva) | 1987 | 1212 |
| Improvement of irrigation in the area of Guadix (Granada) | 1987 | 20,075 |
| Improvement of irrigation in the area of Andarax (Almería) | 1988 | 11,823 |
| Improvement of irrigation in the area of Almanzora (Almería) | 1988 | 15,677 |
| Improvement of irrigation in the area of Axarquía-Este (Málaga) | 1988 | 2346 |
| Conversion into irrigated land of the irrigable area of Palma del Río (Córdoba) | 1988 | 3070 |
| Irrigable area of Baza district-Huéscar (Granada) | 1989 | 28,339 |
| Conversion into irrigated land of the area of the River Corbones (Seville) | 1989 | 3805 |
| Improvement of irrigation in the Guaro irrigation system in Periana (Málaga) | 1990 | 6505 |
| Improvement of irrigation in Contraviesa (Granada) | 1990 | 1576 |
| Agrarian reform measures in the area of Poniente (Almería) | 1990 | 99,700 |
| Measures in this area [in terms of agrarian reform in the Campiña area of Cádiz] and conversion into irrigated land of the irrigable area of Villamartín (Cádiz) | 1991 | 3318 |
| Total | | 2,033,055 |

Note: Source: [19].

The area affected by the irrigation conversion schemes exceeded 200,000 hectares, resulting in thriving agriculture today in some areas, such as western Almería. Additionally, the IARA participated in irrigation schemes previously declared by the State to be "in the national interest", such as the Almonte-Marismas and the Chanza projects, both in the province of Huelva, and subsidies were granted to various irrigation communities and to agricultural companies and entrepreneurs for irrigation modernisation. Therefore, Andalusian agrarian reform was closely connected to "water reform" because of the extension of the irrigation areas and the new crop systems associated with them.

As a result of these conversion policies, the construction of hydraulic infrastructures and the promotion of new socio-economic dynamics, irrigation in Andalusia extended greatly over the last decades of the 20th century, far beyond the historically irrigated areas, which were more plentiful in eastern Andalusia. These new irrigation areas utilised both surface and underground water to primarily benefit intense polytunnel crops, such as horticultural produce and berries, but also fruit cultivation, to the detriment of the classic Mediterranean trilogy of olive groves, cereal crops and vineyards. This is an agricultural sector that creates many jobs and is integrated into international markets; the irrigated area in Andalusia already oscillates, according to official sources, between 941,996 ha [20] and 1,030,804 ha [21]) (Figure 1).

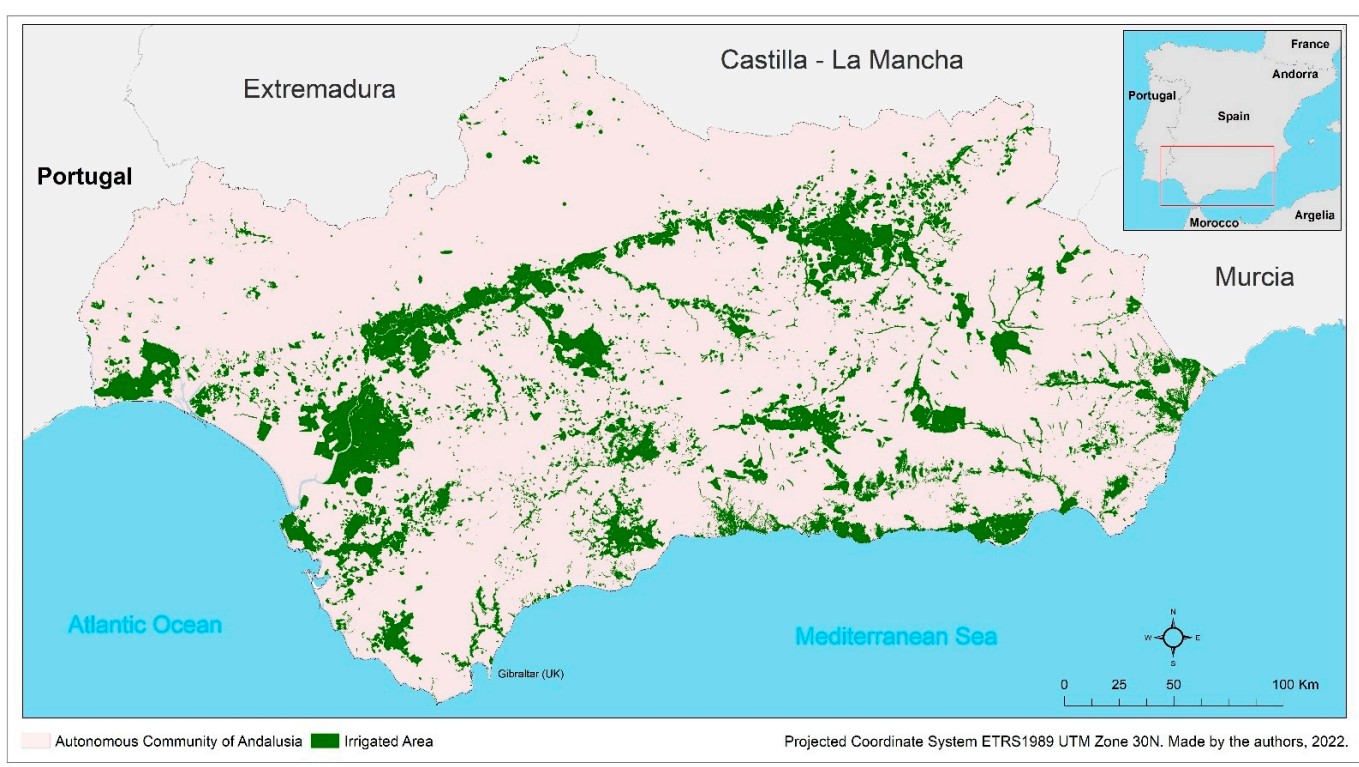

**Figure 1.** Current irrigated area in Andalusia. (Source: [22]).

From 1990 onwards, the regional government's political trust in agrarian reform started to diminish and, with it, the major initiatives entrusted to the IARA, both in terms of irrigation and, mainly, agrarian structure. Andalusian agriculture had gained in technology, production and competitiveness, requiring an active workforce at specific times and, above all, working in increasingly smaller spaces. A large part of the unemployed population of the agrarian sector had been redirected towards other types of employment, guaranteeing the subsistence of those who remained in the agrarian system through rural employment programmes and schemes, which reduced the pressure of day labourers' movements for access to land. In parallel, the agricultural funds and subsidies from the European Union took centre stage in the orientation of agricultural and rural development policies, far

removed from the reformist principles of private property of the Andalusian *latifundia*. Finally, the economic diversification that took place in many rural areas meant a loss of the relative weight of the agricultural sector in Andalusia and, with it, a large share of its social and political influence.

Finally, during the post-2008 economic crisis, the Andalusian Regional Government decided to abolish the IARA in order to absorb its assets and then try to sell them [23]. After this, the process began to dispose of IARA lands by sale in a public auction. However, it failed to produce the success that was expected, as more than a decade later still no buyers have been found for medium-sized and large estates such as: Guzmán I and II (669 ha) and Somontes (394 ha) in Palma del Río (Córdoba); Guadalora (449 ha) in Lora del Río and La Campana (Seville); La Parra (357.5 ha) in Puebla de Don Fadrique (Granada); or the Cortijo de Enmedio (269 ha) in Moclín (Granada). The decree regulating the disposal procedure prioritises the purchase of land by municipalities in the first round [24], but the value of many of the estates exceeds the financial resources available to local authorities and their limited borrowing capacity, which ultimately results in the transfer of public assets to a market in which these properties suffer a continuous devaluation of their appraised value with each failed auction. Nonetheless, part of the lands taken out of the public domain or purchased from private individuals by the IARA in the past are now returning to the hands of a new agricultural entrepreneurship.

In light of this context and with such a historical background, our main objective has been to determine the role played by the body responsible for implementing agrarian reform-the Instituto Andaluz de Reforma Agraria (IARA)-in agrarian transformation. We have focused mainly on the conversion of non-irrigated land into irrigated land, in particular the agriculturally most advanced regions of Andalusia: the irrigable area of the River Chanza on the west coast of the Huelva province.

## 2. Materials and Method

To analyse the role which IARA played in extending irrigation land in Andalusia, from its creation until its disappearance, the historical method has been used [25] with the goal of determining both the temporality of the process, namely to pinpoint the causes and consequences of events and dynamics on the timeline, and its historicity, what means to propose an interrelated, coherent and verifiable narrative of the key role of the IARA in the agricultural transformation of the west coast of Huelva. To this end, we have focused in particular on the content analysis of documentary sources [26,27] as technical documents from the Central Archive of the CAP; legal/regulatory documents from the Official Gazettes of Andalusia and the Spanish State; and agro-statistical documentation published by the Andalusian Regional Government in its specialised databases (Andalusian Multiterritorial Information System, Andalusian Institute of Statistics and Cartography, CAP Agricultural Structures and Infrastructures Service, CAP Water and Coastal Service and Andalusian Chamber of Accounts Documentation and Communication Service). After locating the most relevant documentation and sorting it by source, issuing body, date of creation, nature of the content, subject matter, a time frame of validity and geographical scope, we proceeded to analyse the content following a mixed strategy, i.e., operationalising on the one hand key measurement variables for geostatistical data and, on the other, variables of socio-historical significance for qualitative data. The fundamental agro-statistical variables were location, surface area, volume, capacity and potential of irrigation and of crop changes in land use. The qualitative variables, which were ordinary, were the timeframes, policy decisions, regulatory changes and socio-economic dynamics associated with the creation of the IARA and irrigation extension in Andalusia from the Transition to Democracy (the mid-1970s to mid-1980s) to the present day. Finally, the interpretation of the results was triangulated by integrating quantitative and qualitative data and checked against existing literature on Andalusian land reform, whose most outstanding production is from the mid-eighties to the mid-nineties [28–34].

Despite its 27 years of existence and a multi-million-euro investment in public works and agro-economic intervention, it should be noted that the IARA and the LRAA have been poorly supported by the information and publication services of the Andalusian Regional Government. With few exceptions, such as the chapter on this subject by Santiago Bujalance in his work "*Historia de la agricultura andaluza*" [History of Agriculture in Andalusia] [35], no works summarise or analyse in depth the initiatives of the first and last Andalusian agrarian reform with the benefit of hindsight. The current CAP has custody of part of the agricultural action and development files, thanks to the professionalism of its archivists; however, there is a significant dispersion of documentation in the IARA in the provincial delegations, and there are questionable oversights in the chain of custody of documents, such as the situation in Huelva where documentation ended up rotting in the wine cellars of a woodland village in the Doñana National Park.

Despite the vast developments in statistics that have taken place in recent decades in terms of statistical and mapping production and dissemination in Andalusia, public bodies such as the IECA have no statistics on the IARA, nor on the Research and Statistics Department of the CAP itself. In this respect, of the various statistical works produced and disseminated by this Department since the beginning of the nineties (statistical yearbooks, regional accounts, monthly bulletins, etc.) the only records of the actions of the IARA are to be found in its old yearbooks "*La agricultura y la pesca en Andalucía*" [Agriculture and Fishing in Andalusia] [21]. These reports, published between 1985 and 2009, dedicate a chapter to "agrarian structure policies", which provides insight, albeit in a very summarised form, into the yearly workings of the IARA. The first reports, from the mid-eighties, provide information on specific actions, while the latest reports cite only the IARA in terms of "the management of its assets", already in the process of being disposed of.

Despite the Administration's evident debt with the inventory and dissemination of data related to agrarian reform, the data obtained and analysed during the study allows us to show, as set out below, the key role of the IARA in the conversion of large areas of non-irrigated land into irrigation land, focusing in particular on the coast of the Huelva province and in particular on the 17,200 hectares of the irrigation perimeter of the Chanza.

## 3. The Territorial Framework. Irrigation and Its Landscapes on the Coast of Huelva

The province of Huelva was characterised by the almost non-existence of historical irrigation systems. Until the late 20th century, it was still the province in Andalusia with the fewest irrigation areas. In 1916, it scarcely represented 1% of irrigation land in Andalusia, a figure that rose during the second half of the 20th century to 4% in 1997 [36] and saw a slight increase to 4.57% in 2020 [22]. The IARA participated in this extension by collaborating in the conversion of three irrigable areas: two on the eastern coast of Huelva (Valdemaría and Almonte-Marismas) and one on the western coast (Chanza).

Expansion in irrigation land in Huelva first started in the mid-eighties and was linked to the private initiative of the farmers who embarked on the enterprise of adapting their fields to the newly emerging strawberry cultivation-irrigated agriculture producing very favourable economic results for the pioneers of that "red gold" [37,38]. The presence of surface water and, above all, groundwater, together with bank loans, assisted the development of this water-demanding crop, which not only transformed cultivated areas but also technology in the fields, and led to the creation of cooperatives, marketing and transport structures of international scope hitherto unknown in the area. Furthermore, it resulted in an increased presence of temporary workers from neighbouring countries, such as the north of Africa, and further afield, such as Eastern Europe and Latin America.

In just a few years, agriculture that was hitherto marginal became the economic engine of these coastal areas, reflected by a generation of income and public-private investment, an increase in paid employment both in the countryside and in manufacturing plants, the emergence of auxiliary industries, an increase in commercial activity, the provision of new infrastructures, demographic growth and urban expansion, etc. Strawberry agriculture

showed considerable leverage in favour of processing, marketing and transport sectors, and also removed the threat of population decline in these areas.

Along with the progressive transformation of the economic and social structure of the area, there has also been a substantial change in the agricultural landscape: the decline in forest land and rainfed crops has been directly proportional to the increase in irrigated land, resulting in an atypical landscape of plasticulture and polytunnels, known today as "seas of plastic". Thus, in 1981, the irrigated land of the west coast area, the eight municipalities from the River Guadiana to the River Odiel, barely covered 2200 ha, 2.1% of total surface area and 6.6% of arable land. In the same area, in 1989, there was already an approximate total of 9100 ha (8.7% and 27.1%, respectively). Expansion continued and, today, much of the countryside has been taken over by irrigated agriculture, leaving the non-irrigated arable land practically on the sidelines. The latest statistics from the IECA (2021) [22] on general land distribution in 2020 show 16,938 ha of land for irrigation purposes—16.2% of total surface area and 58.72% of arable land. Within this irrigated area, the most important crops are now woody crops (15,427 ha) compared to herbaceous crops (1511 ha).

On the eastern coast, a similar process occurred in the agricultural areas of Moguer-Palos and Almonte-Doñana, although here herbaceous crops, mainly represented by the berry family, have continued to be in the majority. In this area, forest land converted into strawberry fields has not always been privately owned. Municipal pine forests cultivated in sandy areas at a distance from agricultural lands suffered on many occasions encroachment, logging, conversion of use and unlawful groundwater irrigation.

Therefore, the province of Huelva has three initial sites of irrigation extension: two on the east coast (Moguer-Palos and Almonte-Marismas) and one on the west coast (La Redondela-Lepe-Cartaya), followed by areas of further inland expansion (Andévalo Occidental and the Cuenca Minera).

As for the source of the water, although initially all irrigation was fed from the subsoil, identifying the source of the water (Aquifer 25 on the west coast and Aquifer 27 on the east) was not possible. Soon, major hydraulic works began, leading to the widespread use of surface water irrigation along the entire west coast, in addition to new irrigation areas inland. In contrast, irrigation areas in the east are still fed mainly by underground water. It continues to be an unresolved problem, with deep socio-political and environmental repercussions, as it pits farmers against each other and against other economic sectors for scarce water, depleting the aquifer that feeds the protected natural areas of Doñana [39–41]. Now, only irrigation in the municipality of Palos de la Frontera and a part of the municipality of Moguer comes from surface water, specifically from the Chanza-Piedras System.

Focusing once again on the west coast, in the early eighties, the extension and conversion of irrigation land depended on the groundwater quality and groundwater exploration (Aquifer 25), as well as the agrological quality of land and, especially, personal initiative and the farmers' access to credit. Even then, modern orange plantations were established by companies with outside capital (known locally as the *valencianos*, because of the Valencian origin of many of their owners), exceeding the possibilities of family economies in the area. Orange plantations were established on sandy and clay soils used previously for pine and eucalyptus forestry after considerable clearing, terracing and soil correction works. On the other hand, small agricultural holdings were more suited to the *fresón* (large strawberry), which was more likely to make a return on investment and reap profits in the short space of a season (one year) and, at the same time, greater use could be made of family labour.

The progression of these crops and their impact on the aquifer in question, due to overexploitation and deficient water quality, led to a request from the sector for the public administration to create other forms of provisioning. This resulted in the construction of the Chanza Reservoir, whose water resources connected with the Piedras Reservoir (65 hm$^3$), which, since its construction in 1968, had covered urban and industrial needs but had failed to meet agricultural demands. The Chanza Reservoir was opened in 1989, capturing the waters of this large tributary right at its mouth in the main course of the River Guadiana.

Its construction and the use of its waters were exclusively for Spanish interests under the Spanish-Portuguese Agreement of 1968 for the use of international rivers. Later on, in 1998, this Agreement was transformed into the Agreement on Cooperation for the Protection and Sustainable Use of Spanish-Portuguese Waters and Hydrographic Basins, also known as the Albufeira Convention [42].

Other works, such as the San Silvestre de Guzmán Tunnel and the El Granado Canal, joined the Chanza with the Piedras Reservoir, whose waters connected with the capital of the Huelva province through the Piedras Canal. Added to these resources are the Bocachanza Pumping Station, which takes some 35 hm$^3$ directly from the Guadiana. The Pumping Station is at the foot of the Chanza Dam and drives fresh water from the River Guadiana to the El Granado Canal, with over 135 metres of gradient. One last reservoir, the Machos (1988), of 12 hm$^3$, collects water from this system to bring it to the western coast for agricultural and urban uses. These infrastructures form the main water system of the province of Huelva: the Chanza-Piedras System.

Finally, the impulse of the hydraulic infrastructures went hand in hand with the emergence of modern irrigation communities, which became the farmers' advocates when dealing with the different public administrations. One of these first institutions, which arose in the wake of the new agriculture and with a view to managing the future surface water of the Piedras and Chanza dams, was the Comunidad de Regantes de Lepe (Lepe Irrigation Community), created in 1983.

## 4. Results. Irrigations in the Chanza Area

### 4.1. Background

It became the main project in a set of irrigation schemes that administrations such as the IARA and the emerging irrigation communities created for the province of Huelva. Table 2 presents the irrigation forecasts in 1989 on the basis of the projections dictated by the hydrological scheme of the time, the Guadiana Hydrological Plan.

**Table 2.** Irrigation zone of the south of the Huelva province.

| Irrigation Schemes | Irrigable Area (ha) |
| --- | --- |
| Chanza | 17,272 |
| Sur-Andévalo | 11,049 |
| Moguer-Palos-Lucena (Irrigable area of Valdemaría) | 4180 |
| Total area to be irrigated with water from the Chanza system | 32,501 |
| Corumbel (La Palma del Condado) | 1451 |
| Lucena del Puerto | 500 |
| El Corunjoso (Paterna, Escacena, Manzanilla and Villalba del Alcor) | 3333 |
| Total area to be irrigated with water from the Tinto System | 5284 |
| Irrigable area of the Beas Dam | 810 |
| Irrigable area of Almonte-Marismas | 6700 |
| TOTAL OF SOUTH OF PROVINCE | 45,295 |

Note: Source: [43,44].

In this context, the commission entrusted with promoting the irrigable areas of the province of Huelva (Comisión Promotora de las zonas regables de la provincia de Huelva-COPREHU) provided for an increase in surface water storage capacity by 1990 in anticipation of the hydraulic works in the pipeline, including groundwater back then. A storage volume of 1797 hm$^3$ and a regulation capacity of 912 hm$^3$ were estimated, compared to the actual 742 hm$^3$ and 361 hm$^3$, respectively, at that time.

Undoubtedly, the main goal in the planning undertaken in the Chanza Irrigation Project consisted in mitigating the risk of danger of overexploitation of the aquifer, while ensuring the quantity and quality of, water supply for the expansion of the irrigated area.

As a result, by the Royal Decree 1242/1985 of 17 July, the current irrigable area of the Chanza [45], whose conversion into irrigation land was declared to be "in the national

interest", was constituted. The explanatory memorandum already indicated the strong demand from the agricultural sector and the insufficient water supply and referred to the works on the Chanza that were being carried out at that time by the then Ministry of Public Works and Town Planning (Ministerio de Obras Públicas y Urbanismo-MOPU).

The Royal Decree states that the Ministry of Agriculture, Fisheries and Food (Ministerio de Agricultura, Pesca y Alimentación-MAPA) had already carried out agrological studies of the soils and the profitability of the irrigation, creating an initial irrigable area of 13,500 ha out of a total area of 21,500 ha in the municipalities of Lepe, Cartaya, Aljaraque, Gibraleón, Punta Umbría, Villablanca, Isla Cristina and Ayamonte. Years later, the irrigable area was extended to 17,275 ha.

Although this scheme was declared to be of national interest, the Royal Decree already highlighted the leading role played by the newly formed Andalusian Regional Government (Junta de Andalucía). In this regard, the Agreement on the Joint Committee on Transfers of 1 February 1984 was hereby approved on the transfers of functions and services of the State Administration in the field of agrarian reform and development, approved in the Royal Decree 1129/1984, of 4 April [13], drafting a proposal for collaboration in carrying out studies and fieldwork, as well as the financing and implementation of actions.

Reference was also made to a previous declaration regarding the irrigable area of the Chanza, stating 4000 hectares of useful irrigable area, which was also declared to be of national interest by the Royal Decree 2893/1982 [46]. Therefore, this area was integrated into the new planning of the irrigable area along with the rest of the new area to all intents and purposes (from 1985) with the aim of carrying out a global and uniform action throughout. This initial irrigable area extended to the municipalities of Lepe, Cartaya, Aljaraque, Gibraleón and Punta Umbría. The technical development of this first scheme was entirely in the hands of the National Institute for Agrarian Reform and Development (Instituto Nacional de Reforma y Desarrollo Agrario –IRYDA-), to whom the General Transformation Plan for the irrigation area was entrusted, in the manner laid down in Article 97 of the above-mentioned Law on Agrarian Reform and Development (Ley de Reforma y Desarrollo Agrario) [47].

Therefore, with the new Royal Decree 1242/1985 [45], the irrigable area was extended to the municipalities of Villablanca, Ayamonte and Isla Cristina. Art. 3 states that "the planning actions for the conversion of the area will basically consist of irrigation conversion, redistribution of property, promotion of holdings with appropriate socio-economic characteristics and improvement of the rural environment, to be implemented in the form of a General Transformation Plan." Further on, the regulation states that "the execution by the Autonomous Community of the actions that correspond to it ( . . . ) may be implemented in accordance with the Law 8/1984, of 3 July [14], on Agrarian Reform in Andalusia and its regulation, approved by Decree 276/1984, of 30 October [17]".

In this regard, although the scheme was declared to be of national interest and stemmed originally from the State initiative, the regional administration was responsible for the studies and work that were to serve as a basis for the drafting of the Conversion Scheme, although maximum collaboration and coordination between the two administrations were called for (Art. 4.1.). Finally, Art. 4.2. states that "it is the National Institute for Agrarian Reform and Development (Instituto Nacional de Reforma y Desarrollo Agrario –IRYDA-), subject to an appropriate proposal for cooperation from the Autonomous Community in order to finance and carry out actions, who, jointly with the Andalusian Institute for Agrarian Reform (IARA), shall draw up the General Transformation Plan, which will be approved by a Royal Decree", which, logically, will be later.

*4.2. Operation*

Given the magnitude of the scheme, both organisms agreed to subdivide the project into two areas using the course of the River Piedras as a differentiator. The result of this coordinated work led to early official approval. In July 1986, the General Conversion

Scheme for the first part or East Sub-area was passed [47]. A year later, in July 1987, the General Transformation Plan for the second part or West Sub-area was passed [48].

In both Royal Decrees, and for each scheme or sub-area, the irrigation systems to be adopted and the works necessary for the conversion of the area were specified, as well as the production guidelines, aids and incentives to be granted to agricultural holdings, establishing which holdings were to be considered as standard farms, as well as the actions to be encouraged to improve the countryside. The participation of the State and regional administrations in its implementation was also highlighted.

The territorial area concerned was divided into two sub-areas and these, in turn, into sectors. In terms of the phases of implementation of the project, there are three phases (Tables 3 and 4 and Figure 2). The West Sub-area is located between the Guadiana and Piedras rivers, with a net irrigable area of 12,337 ha, spread across the municipalities of Lepe, Isla Cristina, Villablanca and Ayamonte. The East Sub-area (4935 ha), between the Piedras and Odiel rivers, covers the municipalities of Cartaya, Gibraleón, Aljaraque and Punta Umbría.

**Table 3.** Phases and sub-areas of the Chanza irrigable area.

|  | Irrigable Area (ha) | Net irrigable Area (ha) |
| --- | --- | --- |
| EAST sub-area | 6991 | 4935 |
| WEST sub-area | 16,155 | 12,337 |
| TOTAL | 23,146 | 17,272 |

Note: Source: [43,44].

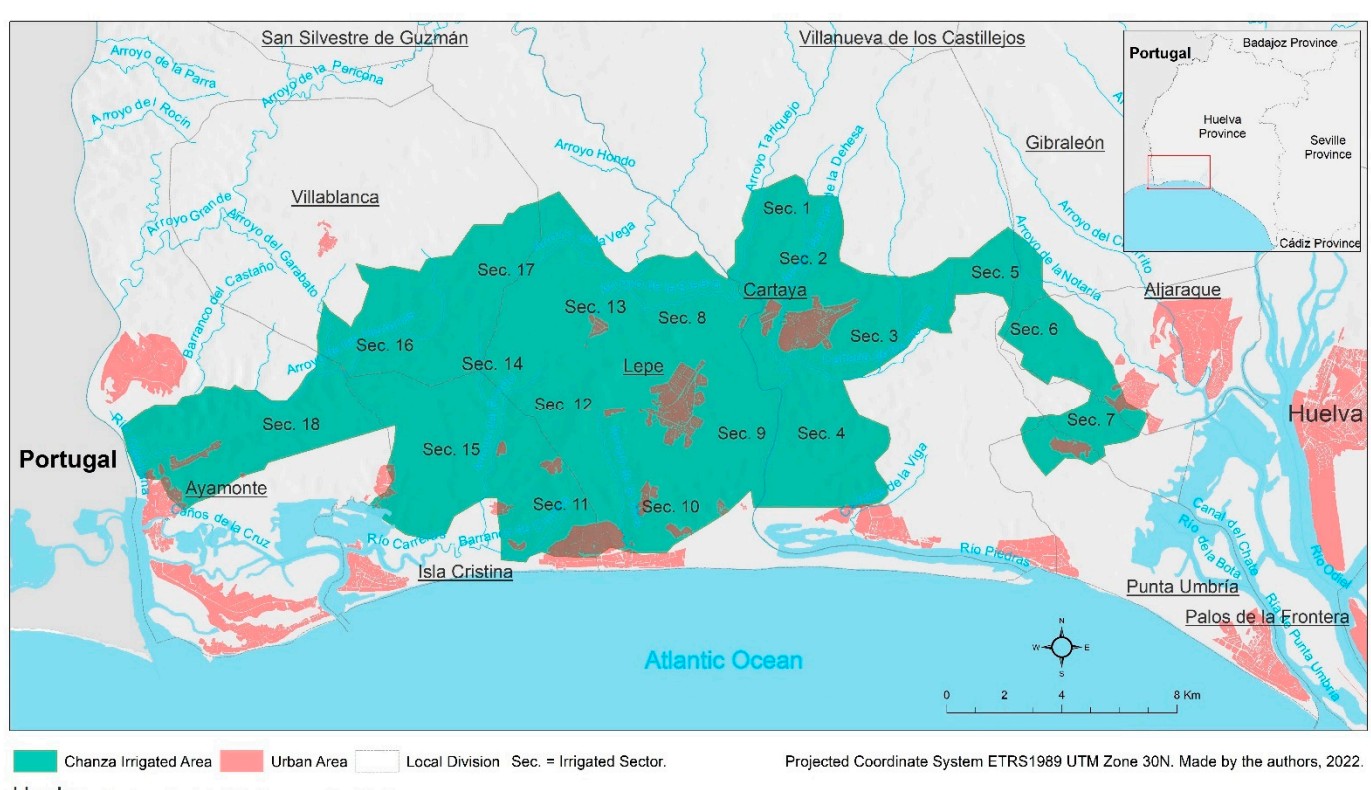

**Figure 2.** Chanza Irrigable Area. (Source: [49]).

**Table 4.** Phases, sub-areas and sectors of the Chanza irrigable area.

| Phase I or East Sub-Area | | | |
|---|---|---|---|
| Sector | Name | Town | Net Irrigable Area (ha) |
| 1 | Tariquejo | Cartaya | 888 |
| 2 | Cartaya-Norte | Cartaya | 584 |
| 3 | Cartaya-Sur | Cartaya | 687 |
| 4 | Garranchal | Cartaya | 998 |
| 5 | Mogayuela | Cartaya | 812 |
| 6 | Aljaraque-Norte | Aljaraque-Gibraleón | 382 |
| 7 | Aljaraque-Sur | Aljaraque-P. Umbría | 584 |
| | **Total Phase 1** | | **4935** |
| | Phase II or West Sub-Area | | |
| 8 | Cabezarias | Lepe | 1131 |
| 9 | Río Piedras | Lepe | 880 |
| 10 | La Antilla | Lepe | 785 |
| 11 | La Redondela | Lepe-Isla Cristina | 1837 |
| | **Total Fase 2** | | **4633** |
| | Phase III or West Sub-Area | | |
| 12 | La Tejita | Lepe | 1131 |
| 13 | Mesa del Turmán | Lepe | 1364 |
| 14 | Cañada del Galgo | Lepe-Isla Cristina-Villablanca | 1164 |
| 15 | Carrasquito | Isla Cristina | 1058 |
| 16 | El Marquesado | Villablanca-Isla Cristina-Ayamonte | 1009 |
| 17 | Villablanca | Villablanca-Lepe | 727 |
| 18 | Ayamonte | Ayamonte | 1039 |
| | **Total Phase 3** | | **7704** |
| | **Total Irrigable Area of the Chanza** | | **17,272** |

Note: Source: [43,44].

For hydraulic infrastructure purposes, each sector was subdivided into plots of 20–25 ha. They were all fed by a water intake from the main canal, which in turn had a subsequent irrigation system to reach each agricultural holding.

The area's conversion works were subdivided among the various bodies and private individuals. Therefore, the then Ministry of Public Works and Urban Development (MOPU) became responsible for the basic irrigation water supply network and regulating reservoir, road and stream infrastructure works. Accordingly, the IARA was responsible for (a) works of general interest: road network, drainage network and electrification; (b) work of common interest: pumping stations, common irrigation distribution network. There are also, finally, works of private agricultural interest, i.e., between the general intakes to the farms and the irrigation networks to the holdings. A Joint Technical Commission was set up, consisting of three representatives from the IRYDA, three from the MOPU and six from the Autonomous Community of Andalusia. The scheme had to be drawn up within one year, and could be developed in several phases in order to coordinate the works with the other conversion works (Art. 5 and 6 of the Royal Decrees 1411/86 [47] and 876/1987 [48]).

In determining the area unsuitable for irrigation, physical criteria (presence of rocky areas, marshes, steep slopes, etc.) and ecological criteria (preservation of stone pine and cork oak forests and other legally protected areas, etc.) were followed.

These works involved a considerable amount of money. As for financing, the works considered to be of common interest were subsidised at 40% non-repayable, while those of private interest, although managed by the emerging irrigation communities, were also subsidised on long-term loans at a low interest.

This Chanza Project is the main reason why the irrigation communities were created. They were responsible for planning hydraulic resources with a view to meeting water demand and rationalising uses and activities in an environmentally friendly way.

The works of general interest were developed and completed quicker in the East Sub-area. In the West Sub-area, in light of the delays, irrigation was developed under precarious conditions, whereby the canalisation works were paid for by the interested parties, despite the cost overruns. This is proof of the high demand and power of this new agriculture.

*4.3. Socio-Economic Consequences in the Area*

From the beginning, it was clear that this irrigation project would have an enormous magnitude in this region and that it would be justified by an increase in agricultural production, growth in the labour market and the direct and indirect benefits for the productive fabric. The total investments made (reservoirs, canals, roads, electrification, pumping and irrigation distribution network) were expected to have an impact on:

(a)  Increasing the final agrarian production.
(b)  Creating several thousand jobs, some permanent and others, indirect employment, with a significant transfer of local labour and capital from other sectors of activity to the agricultural sector. It had already been envisaged that this demand would be met by increasing migrant labour, which, years later, became immigrant labour.
(c)  Promoting crop diversification, avoiding the risks of monoculture, as strawberries almost monopolised the new irrigated crops.
(d)  Increasing yields obtained through surface irrigation and new irrigation techniques applied.
(e)  Avoiding overexploitation of the aquifer. Since then, groundwater extraction has been prohibited.

In short, this General Transformation Plan for the Chanza Irrigable Area became an essential element in the development of the agricultural activity. Firstly, by guaranteeing an adequate supply of surface water in terms of quantity and quality, thus avoiding the extraction of groundwater and, secondly, by extending the irrigable area, with all the socio-economic consequences that this entailed.

Therefore, the agrarian and productive changes that were already on the horizon in this coastal region were intensified. As a result, the expansion of irrigation made it possible to extend and improve transport and marketing channels (cooperatives and private companies), while cultivation techniques that increased yields emerged, industries, businesses, services (agribusinesses) and urban land for industrial use multiplied, the workforce in fields and warehouses grew and farm incomes rose. In just a few years, agriculture went from being a marginal activity that drove its farmers out to an attractive activity for them and their children.

Many IARA forecasts have been fulfilled. Therefore, significant growth has been seen in production and employment in this part of the west coast, and this is also applicable to the entire coastline. In Section 3 above, we have already referred to the current 16,938 ha of irrigation land.

With regard to agricultural labour, expectations have been exceeded. Local or subregional labour is no longer sufficient. Manpower, consequently, has grown at the same rate as the irrigation area. In the 2021 berry season (strawberries, raspberries, blackberries and blueberries), for the whole Huelva province over 100,000 workers were employed, of which

50,489 labourers were foreign nationals [50]. Therefore, for the last two decades, the west and east coasts of Huelva have been a testing ground for what is known as recruitment at the origin. Years ago, primarily Moroccan nationals were recruited (12,000 in the 2020–2021 season, mainly women), but since the recent strawberry season of 2021–2022, Honduran and Ecuadorian nationals (around 500 workers) have also been recruited in their home countries.

## 5. The Progressive Expansion of Irrigation in the Rest of the Huelva Province

In light of the experience of the irrigation areas of the Chanza, the water availability of the Chanza-Piedras System and the development of new cultivation techniques have extended the agricultural frontier to new areas that, until recently, were inconceivable, especially in a land that was traditionally forest land. This has also been a result of the construction of new reservoirs that have increased the overall water resources of the province of Huelva (Table 5).

**Table 5.** Reservoir capacity of the Huelva province.

| | Capacity (hm$^3$) | Area (ha) | Year of Construction | Main Water Uses | Municipalities Where Located | Type of Reservoir |
|---|---|---|---|---|---|---|
| **Tinto-Odiel-Piedras Basin (Huelva)** | | | | | | |
| CHANZA | 338 | 2239 | 1987 | Irrigation and urban water supply | El Granado | Gravity |
| PIEDRAS | 60 | 796 | 1968 | Irrigation and urban water supply | Lepe-Cartaya | Rock fill. Concrete |
| LOS MACHOS | 12 | 182 | 1988 | Agricultural irrigation | Lepe-Cartaya | Rock fill. Concrete |
| ANDÉVALO | 634 | 3500 | 2004 | Irrigation and urban water supply | El Granado-Puebla de Guzmán | Gravity |
| CORUMBEL BAJO | 19 | 396 | 1987 | Irrigation and urban water supply | Palma del Condado | Rock fill. Concrete |
| BEAS | 3 | 36 | 1927 | Urban water supply | Beas | Gravity |
| EL SANCHO | 58 | 427 | 1962 | Industrial | Gibraleón | Gravity |
| JARRAMA | 43 | 342 | 1999 | Irrigation and urban water supply | Nerva | Rock fill. Clay core |
| ODIEL | 8 | 78 | 1970 | Irrigation and industrial use | Aracena | Rock fill. Sheet pile |
| OLIVARGAS | 29 | 240 | 1982 | Irrigation and industrial use | Almonaster la Real | Gravity |
| **Hydrographic Confederation of the River Guadalquivir (Huelva)** | | | | | | |
| ARACENA | 127 | 844 | 1969 | Urban water supply | Puerto Moral, Aracena and Zufre | Buttress |
| ZUFRE | 179 | 943 | 1991 | Urban water supply | Zufre | Rock fill. Clay core |
| **TOTAL** | 1.510 | 9.987 | | | | |

Note: Source: [51]. Elaboration by authors, 2022.

These include the construction of the Andévalo Reservoir (634 hm$^3$). Completed in 2003, the works provided greater water resources to a water system consisting of the river basins of the west of the province of Huelva (Figure 3), the Andévalo-Chanza-Piedras System [52,53]. This system has become a kind of "River Nile" for the Huelva province, given its economic importance. It is a set of water transfers, canalisations and pipelines that takes the water to the crop fields (Figure 4). Precisely, one of the main demands of the agricultural sector today is to fulfil the initial expectations of this system, which is to improve the water distribution network and connect with the Odiel system up to Doñana (Guadalquivir Basin).

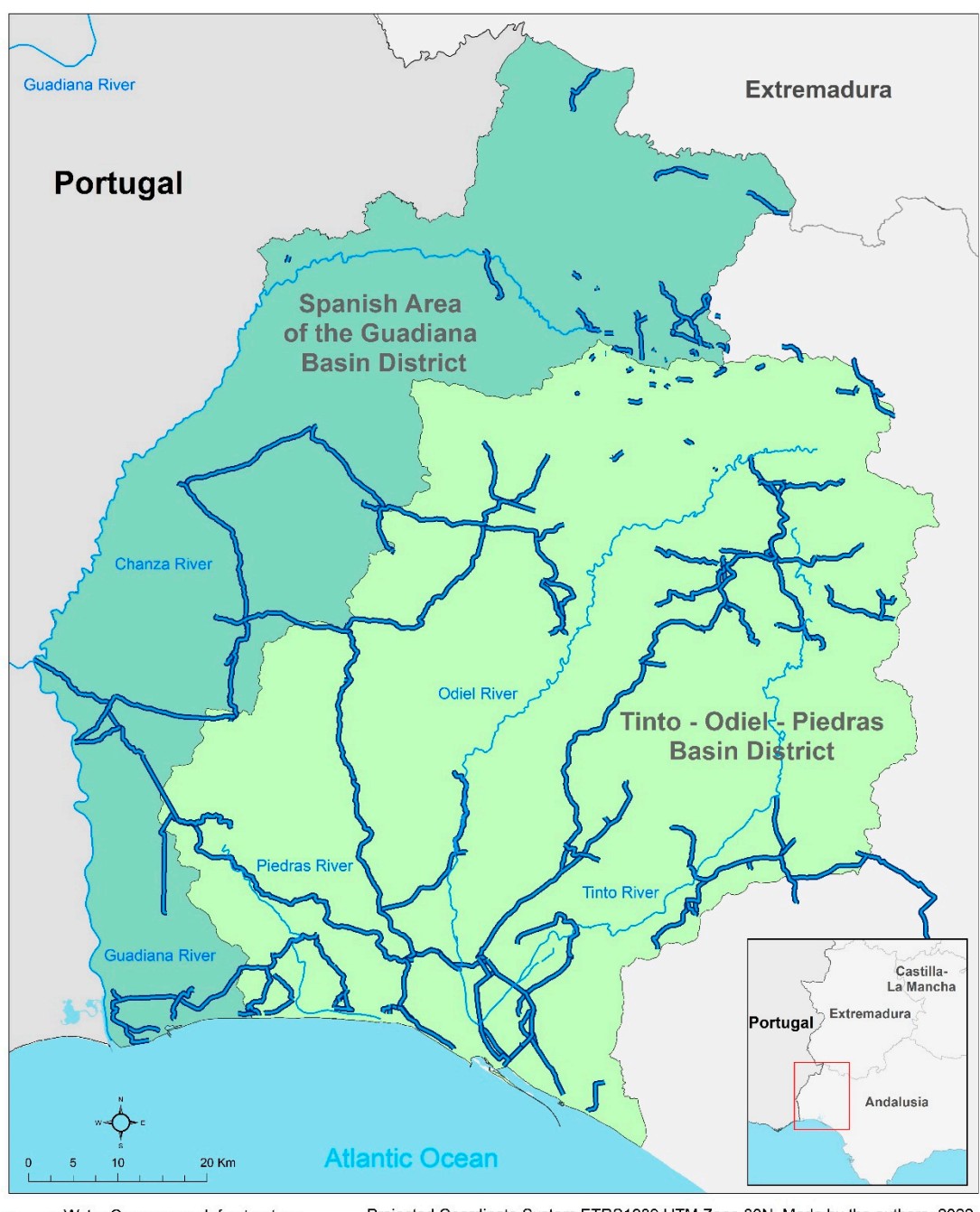

**Figure 3.** Water conveyance system originating in the intra-community river basin districts of Guadiana and Tinto-Odiel-Piedras. (Source: [54]).

As a result, irrigation is expanding further north from the coast, in the western Andévalo area, in soils that are mostly sand, gravel and slate. These are forest lands, but with the new irrigation techniques and the availability of water, they are converted to accommodate citrus and a variety of other water-demanding fruit crops, such as avocados.

This is how new schemes surface (and with them, irrigation communities), declared to be of regional interest, but which were also originally developed by the IARA. These include the Sur-Andévalo Irrigation Scheme, extending north of Cartaya, Villanueva de los Castillejos, Gibraleón and San Bartolomé de la Torre, managed by the Sur-Andévalo Irrigation Community. In turn, further north, participating municipalities from the irrigation communities of Andévalo-El Almendro, Andévalo Fronterizo and Andévalo Minero

have also been trying for a few years to expand irrigation on their forest lands as a formula for local development. In the two first, direct use is made of the water from the Andévalo Reservoir, and the Andévalo Minero uses water from the Olivargas Reservoir (River Odiel). Likewise, in Villanueva de los Castillejos, the Pedro Arco Irrigation Community, from the García Carrión company, has been created, and it obtains water from the exit of the tunnel of San Silvestre. The same intake supplies the farms of the Andévalo-Guadiana Irrigation Community in Villablanca and San Silvestre.

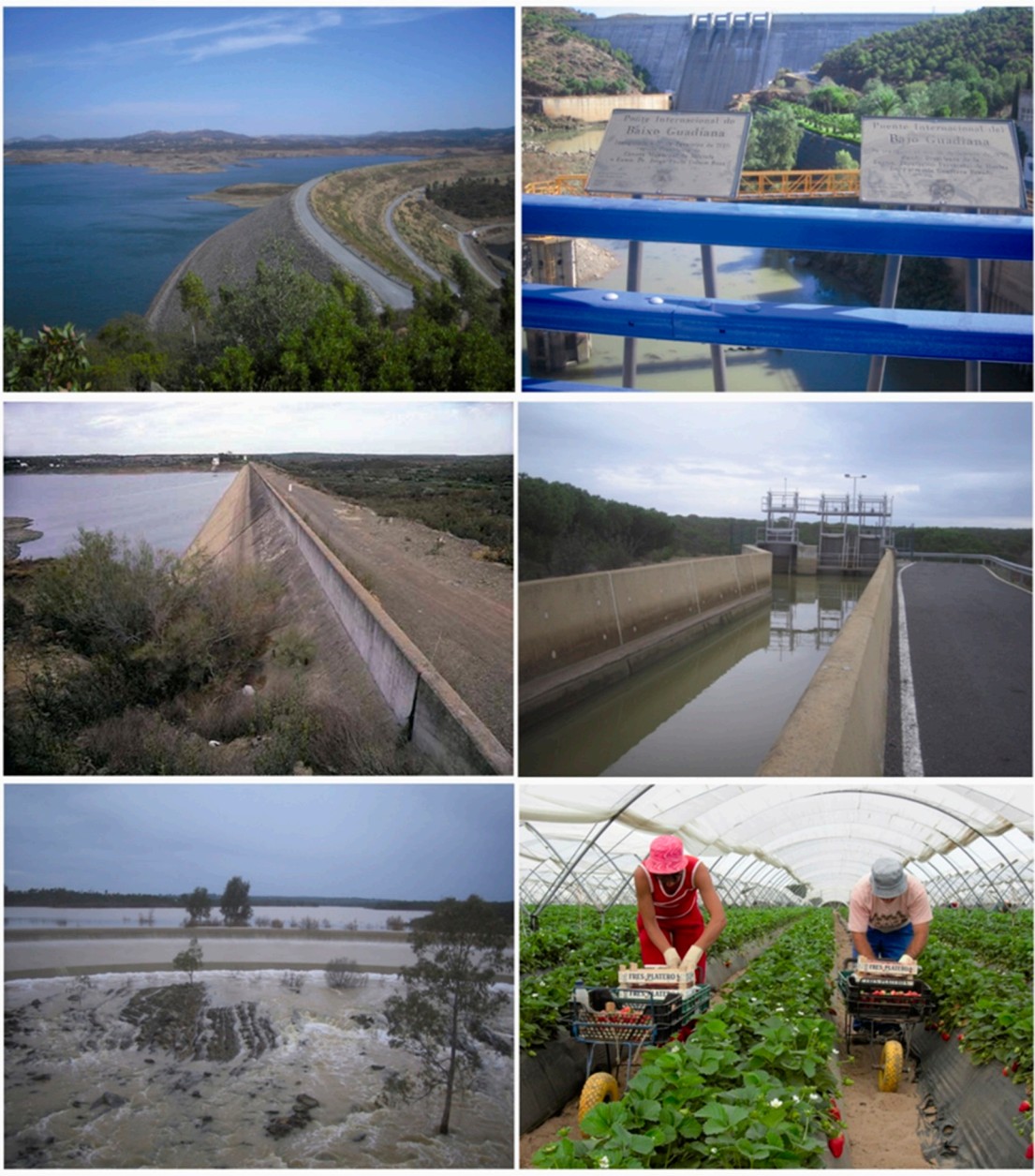

**Figure 4.** Hydraulic infrastructures of the Andévalo–Chanza–Piedras–Machos System and strawberry fields in Huelva. (Source: [52,53]).

Likewise, the number of irrigation communities has multiplied (Table 6), in contrast to other provinces with long traditions in terms of irrigation, when historically in the province of Huelva these communities did not exist. Precisely the Chanza Irrigation Project gave rise to the irrigation communities of Piedras-Guadiana, located in Lepe (1983), and Chanza-Piedras in Cartaya (1988), which were the first irrigation communities to be created

and from whose experience the others were forged. They became corporations of public law with delegated powers in water matters by the Andalusian Regional Government (since 2005). At present, farmers' petitions to administrations responsible for water, agriculture and the environment are at the top of the list. They have, in turn, come together in a larger entity: the irrigation community "Comunidades de Regantes de Huelva" (COREHU).

**Table 6.** Irrigation Communities in the River Basin District of Huelva (1).

| Affiliated IC | Date of Inscription of the IC | No. of Applicant Irrigators | Potentially Irrigable Area | Use (hm$^3$) 2015 | Source of Current Supply |
|---|---|---|---|---|---|
| C.R. Andévalo- El Almendro | 16/11/2004 | 5 | 570 | 1248 | Andévalo |
| C.R. Andévalo-Pedro Arco | 03/12/2004 | 3 | 1500 | 7508 | Chanza System |
| C.R. Andévalo Fronterizo | 10/11/2004 | 180 | 10,000 | - | Andévalo |
| C.R. Andévalo Minero | 29/12/2005 | 52 | 2836 | - | Olivargas |
| C.R. Andévalo-Guadiana | 02/08/2000 | 145 | 3500 | 6129 | Chanza System |
| C.R. Chanza-Piedras (2) | 21/04/1988 | 2297 | 9022 | 19,506 | Chanza System |
| C.R. Sur-Andévalo | 04/11/1992 | 648 | 10,275 | 30,434 | Chanza System |
| C.R. Corumbel-Corunjoso | 04/12/1998 | 215 | 1542 | - | Corumbel |
| C.R. El Fresno | 31/08/2001 | 400 | 3811 | 12,593 | Chanza System |
| C.R. Onuba (Gibraleón-Aljaraque) | 17/01/2017 | 11 | 1570 | 7211 | Chanza System |
| C.R. Palos de la Frontera | 15/06/2000 | 425 | 3500 | 14,571 | Chanza System |
| C.R. Piedras-Guadiana | 17/02/1984 | 1694 | 13,220 | 25,970 | Machos/Chanza |
| C.R. Valdemaría | 24/03/2004 | 267 | 600 | 2758 | Chanza System |
| Other irrigable areas (3) | | | | 7931 | |
| **TOTAL (4)** | | **6342** | **61,074** | **136,960** | |

(1)  These irrigation communities account for the majority of surface irrigation in the province of Huelva and refer to the river basin districts of Tinto-Odiel-Piedras and Encomienda del Chanza.

(2)  This includes the community's own irrigators plus those of the Piedras Canal that are not strictly from this community (5079 hm$^3$).

(3)  The other irrigable areas are Rio Tinto Fruit (7620 hm$^3$) and the holding TriSasur (0.311 hm$^3$).

(4)  For this year the urban consumption was 40.1 hm$^3$ and industrial was 16.5 hm$^3$. Irrigation, therefore, accounts for 70.79% of total regulated water consumption.

Note: Source: [53,55,56].

Hence, there has been a huge increase in hydraulic infrastructures in agriculture in Huelva in recent decades. Attempts are being made to connect the water resources of several subsystems (Guadiana-Andévalo-Chanza-Piedras, Odiel, Tinto and Bajo Guadalquivir Subsystems) into one single system, the Sistema Hídrico Onubense (Huelva Water System). This would secure supply during periods of drought from the Guadiana, traditionally with excess water, to the Guadalquivir, with a higher deficit. However, in recent years, new public investment has been slow to keep up with demand. On the other hand, there are still unresolved resource problems, such as catchments (legal and illegal) in the surroundings of the natural area of Doñana, on the east coast of Huelva.

The landscape, environmental and socio-economic characteristics of the agricultural area of the province of Huelva, especially the coast, have been deeply changed by the extension of irrigation, an effect that is expected to extend to the entire region of the Andévalo when the irrigable area delimited by the hydrological scheme of the Tinto-Odiel-Piedras river basin district, currently under development, shall increase the irrigated area by 50,000 hectares [57].

The water resources available in this province have grown as a result of the different works carried out in recent decades, but there are still long-overdue shortcomings that hinder the extension and conversion of irrigation systems. Without a doubt, many of the goals of the farmers and irrigation communities will not be achieved because of new territorial and environmental scenarios and hydrological planning processes that are slowing

down the construction of some infrastructures such as the case of the Alcolea Reservoir in the River Odiel and which are delaying urgent works such as the extension of the San Silvestre Tunnel. Furthermore, another problem to be solved is water transfer to the area of the Corona Norte in Doñana in order to reduce its dependence on supplies from its depleted aquifer, which would be difficult without the aforementioned Alcolea Dam and sole dependence on the Andévalo-Chanza-Piedras System. Interestingly, the amount of water to be transferred (15 hm$^3$) has been approved, but there is insufficient infrastructure for it to actually arrive. Likewise, the Canal de Trigueros must be built to carry the waters from Alcolea to the northern part of Doñana.

## 6. Conclusions

The goal of this study is to discover the historic significance of the now-defunct IARA in agrarian conversion and commitment to irrigation, in addition to other actions in the field of land structure and agricultural holdings. Since its foundation in 1984, it participated both in the areas declared agrarian reform areas and in many other areas deemed to be of interest for the economic and social development of the Andalusian agricultural sector. It mainly focused on irrigation conversion in the agricultural areas where it acted, taking responsibility for irrigation schemes both in terms of technical execution and in co-financing. This was the only case of agrarian reform in Spain that was passed by the autonomous government of Andalusia, and activity in the territory was maintained for some 25 years. We, therefore, emphasise here the importance of the IARA in agricultural transformation in some areas after only a decade, although largely forgotten since its official demise in 2011.

One of these schemes in which the IARA acted was the Chanza Irrigation Project on the west coast of the province of Huelva. It was initially declared to be "in the national interest", but in the subsequent transfer of competencies, it was the Andalusian Regional Government, specifically, the IARA together with the IRYDA, who took charge of much of the planning and costing. It, therefore, played a key role in this scheme, which had an unquestionable territorial impact as it triggered an unprecedented boost in the economy of the west coastal region of Huelva.

The IARA also responded to the entrepreneurial concerns of farmers and emerging irrigation communities, in our case, on the coast of the Huelva province. The different infrastructure works we refer to as the Andévalo-Chanza-Piedras System transfer surface water from the mountainous area of Sierra de Huelva and the Andévalo, and from the river itself to the coast, definitively solving the limitations imposed by Aquifer 25 in terms of water quantity and quality. At the same time, it allowed for an expansion in irrigation areas, intensifying the fledgling socio-economic phenomena that had been taking place for years. New agriculture that has stimulated economic growth in activity sectors, creating a knock-on effect. Likewise, local populations could not meet the high demand for labour, and this generates and increases intense temporary and permanent immigration processes to this coastline.

Thus, we are witnessing an "agricultural revolution", with an extraordinary increase in irrigation in Huelva: from the coast to the inland areas. This province is undoubtedly where there are still potential water resources to continue extending irrigation areas, unlike other parts of Andalusia or Levante. Nonetheless, new scenarios and questions arise as to whether market demand, always highly variable, can absorb all this production capacity, irrespective of the emergence of environmental problems and regulations that force and/or advise restrictions. Similarly, the real scarcity of labour is already an issue in current agricultural areas and will be a variable to be taken into account for possible and effective irrigation extensions.

Whatever the case, aside from these scenarios for the near future, the main challenge for water policy in Huelva is undoubtedly the need to balance the expansion of irrigation, which is socially demanded, with the preservation of natural environments of unquestionable value, such as Doñana, whose agriculture excessively depletes its groundwater in the absence of the alternative of a surface water transfer that has failed to materialise.

The irrigation infrastructures promoted by the IARA on the Huelva coast have been spreading inland in recent years, creating a new agricultural frontier: the Andévalo area. Irrigation communities are the new protagonists, building the foundations of the local economic model on a commitment to irrigated agriculture. And as irrigation retains the local population, extending irrigated areas is undoubtedly attractive to public authorities who see intensive agriculture as a buffer against local depopulation and population ageing, overshadowing the impacts it has on the landscape and environment.

In a nutshell, water is the source of life in modern agriculture and a primary element that will gain strategic value over the years. Competitiveness, efficiency, water-cost reduction and environmental stability should be some of the premises or challenges of irrigation in the Huelva province today, on which more work needs to be done for a sustainable future, as set out in Goal 6 of the UN's Sustainable Development Goals and the targets of the 2030 Agenda.

**Author Contributions:** All the authors (J.M.J.A. and J.D.D.) contributed equally in the development of the present paper. For the proper paper development, all the phases have been discussed and worked by the authors. All authors have read and agreed to the published version of the manuscript.

**Funding:** The paper is a result of the project PY20_00864 financed by the PAIDI 2020 Plan of the Regional Government of Andalusia and the FEDER Operative Program.

**Conflicts of Interest:** The authors declare no potential conflict of interest with respect to the research, authorship, and/or publication of this article.

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
