# Peer review of "The Role of the Andalusian Institute for Agrarian Reform (IARA) in Irrigation Expansion: The Case of the Chanza Irrigation Project (Huelva, Spain)"

_water, doi:10.3390/w14182931_

Round 1

Reviewer 1 Report

After a first reading of the manuscript, I find interesting information but I miss a brief comparison with similar actions in other countries, or at least within Spain.
The use of literal "translations" of terms from the administrative or political jargon used in Spain will make it difficult for non-Spanish readers to understand.
On the other hand, the manuscript is full of biased partisan political assessments, for me inadmissible in a scientific Journal.

Author Response

Dear Reviewer 1:

First of all, we would like to thank you for the speed in preparing and receiving your review report. We believe that their comments, together with those of the other two reviewers, have led to the initial manuscript being improved in this new version, which is somewhat longer. Next, we point out these improvements:

- To specify the exceptionality of the process described in the study, we have contextualized the Andalusian agrarian reform in the context of Spain (Extremadura region) and some European countries. The text added in chapter 1 of the Introduction is placed, approximately on the 3rd page.

- We have used acronyms more frequently for the purpose of the statement in Spanish of the name of some institutions. We have also proceeded to the English translation in parentheses

On the other hand, we have passed the renewed text through a professional translation.

In conclusion, we trust that these improvements satisfy your observations for publication.

- We have eliminated allusions to national (Adolfo Suárez) and regional (PSOE) governments. We believe that the dates of the events are sufficient.

- We have also eliminated some evaluations such as: “far from the traditional image of the Andalusian landowner”.

- For the rest, we think that there are no subjective assessments by the authors regarding the political events developed around the Agrarian Reform.

Furthermore, we have improved and expanded the text with some comments from the other two reviewers.

In turn, we have passed the renewed text through a professional translation.

In conclusion, we trust that these improvements satisfy your observations for publication.

Thank you and we remain at your disposal

Reviewer 2 Report

This paper details the role of the Andalusian Institute of Agrarian Reform in expanding irrigation, and the logic is clear throughout the article, but there are still some problems in the article.

1.     The pictures in the article should be made clearer, for example in Figures 1 and 2, so that the reader can more clearly distinguish the actual local irrigation situation.

2.     The abstract is just like an introduction. authors need to provide a snapshot of the article. Present abstract very well explained the problem but failed to explain why the reader should read the entire paper. 

3.     In the first part, the authors focus on the establishment and history of IARA, but the subsequent developments after the abolition of IARA should also be analyzed in more detail

4.     In the The territorial framework, the concept of modern irrigation communities should be further elaborated.

5.     In the economic impact section of the article, the authors should have presented the results in more detail, for example, the creation of agricultural production and the increase in employment should have been presented in detail, rather than in general terms. This section should be expanded appropriately to reflect the results of the reforms.

6.     The authors should conclude with a deeper discussion of the implications of Andalusian Institute for Agrarian Reform, which should be discussed at different levels, such as the impact on the environment, local livelihoods, and comparisons with reforms in the agricultural land sector carried out in other regions, for a more in-depth analysis.

This article needs major revision and the author still needs to make appropriate content changes and additions to the article content

Author Response

Dear Reviewer 2:

First of all, we would like to thank you for the speed in preparing and receiving your review report. We believe that their comments, together with those of the other two reviewers, have led to the initial manuscript being improved in this new version, which is somewhat longer. Next, we point out these improvements:

1. Maps 1 and 2 have been modified to clarify the areas and limits of the autonomous community of Andalusia, the province of Huelva, the affected municipalities and the irrigated areas.

2. We have modified the abstract to clearly present the research problem, the main objective, the methodology and a summary of the results and the conclusion.

  1. At the end of the extensive Introduction (last paragraphs) we have referred to the abolition of the IARA in 2011 and, consequently, the sale by auction of multiple properties to City Councils and companies. We have explained that it is a process that to date has not yet been completed. Likewise, in the conclusions we have added a paragraph to that disappearance and forgetfulness of the IARA.
  2. We believe that we have delved into the irrigation communities. However, in the paragraph before table 6, we have added some allusions to their importance. Likewise, we have dedicated information to the irrigation communities in table 6
  3. Achieve, the information on the economic repercussion and on the labor market was until now somewhat generic. However, two long paragraphs with more detailed information have been inserted right at the end of chapter 4.
  4. In chapter 1, around the 3rd page of the original manuscript, a long paragraph has been introduced in which the attempted agrarian reforms that occurred in the region of Extremadura (Spain), Portugal and the then USSR are alluded to. Likewise, in the “conclusions” chapter, mention is made of the singular case of the Agrarian Reform in Andalusia.

Furthermore, we have improved and expanded the text with some comments from the other two reviewers.

In turn, we have passed the renewed text through a professional translation.

In conclusion, we trust that these improvements satisfy your observations for publication.

Thank you and we remain at your disposal

Reviewer 3 Report

An important issue was raised in the manuscript due to the fact that „In the last two decades of the 20th century, the Andalusian Institute for Agrarian Reform (IARA, the acronym in Spanish) played a major role in converting large areas of non-irrigated land into irrigated land in Andalusia, especially in areas targeted for agrarian reform”.

Nevertheless, in order for the manuscript to be published, in the publishing house to which it was submitted, in the opinion of the reviewer, it requires corrections and supplements.

Abstract in the form submitted for review does not meet the requirements set for this part of the scientific study. A few sentences of introducing the research problem, goal, main methodological assumptions and main achievements of the authors within the research goal should be included here.

The Introduction part, introducing the undertaken research problem, should be separated. At the end of this part, there should be a formulated scientific goal of the research and analysis undertaken. Consider using some of the content in Intoduction that is contained in this chapter 1. The IARA and the new irrigated areas in Andalusia. Policy history.

In the opinion of the reviewer, chapter 2, Materials and method, also needs to be supplemented

The authors wrote, with which the reviewer agreed, that the historical method has been used. Justify why, in the opinion of the authors, this method is appropriate for this manuscript and briefly characterize this method. It is true that there is a reference to literature, but in the opinion of the reviewer, this element should be described in the Material and methods section.

According to the reviewer, the monographic method was also used. You should write about it and describe this method, using the relevant literature on the subject in this area.

In general, the Material and methods section should contain a detailed description of the method (s) used, so that it can be replicated by other researchers in relevant studies.

Author Response

Dear Reviewer 3:

First of all, we would like to thank you for the speed in preparing and receiving your review report. We believe that their comments, together with those of the other two reviewers, have led to the initial manuscript being improved in this new version, which is somewhat longer. Next, we point out these improvements:

-              In the methodological section, we have indicated the sense of temporality and historicity of the process (basis of the historical method) and detailed the documentary sources, the selection procedure and documentary organization, the operationalization of the quantitative and qualitative variables, the triangulation and the contrast of results with existing bibliography for the replicability of the study.

-              We have modified the abstract to clearly present the research problem, the main objective, the methodology and a summary of the results and the conclusion.

 - Indeed, we have not made an introduction in which a presentation of the objectives is made, but rather we start directly with the historical, regulatory and bibliographic references of the Agrarian Reform of Andalusia. However, at the end of this chapter 1, and before chapter 2 of Methodology, we have introduced a paragraph that clearly shows the main object of the investigation.

Furthermore, we have improved and expanded the text with some comments from the other two reviewers.

In turn, we have passed the renewed text through a professional translation.

In conclusion, we trust that these improvements satisfy your observations for publication.

Thank you and we remain at your disposal

Round 2

Reviewer 2 Report

The authors carfully responed  all the coments , and I suggest that the journal publish the paper.